# Sample Efficiency Matters: A Benchmark for Practical Molecular Optimization

**Wenhao Gao**[1*], **Tianfan Fu**[2*], **Jimeng Sun**[3,4], **Connor W. Coley**[1,5]

[1]Department of Chemical Engineering, Massachusetts Institute of Technology,
[2]Department of Computational Science and Engineering, Georgia Institute of Technology,
[3] Department of Computer Science, University of Illinois at Urbana-Champaign,
[4] Carle Illinois College of Medicine, University of Illinois at Urbana-Champaign,
[5]Department of Electrical Engineering and Computer Science, Massachusetts Institute of Technology,
[*]Equal Contributions

`{whgao,ccoley}@mit.edu, tfu42@gatech.edu, jimeng@illinois.edu`

## Abstract

Molecular optimization is a fundamental goal in the chemical sciences and is of central interest to drug and material design. In recent years, significant progress has been made in solving challenging problems across various aspects of computational molecular optimizations, emphasizing high validity, diversity, and, most recently, synthesizability. Despite this progress, many papers report results on trivial or self-designed tasks, bringing additional challenges to directly assessing the performance of new methods. Moreover, the sample efficiency of the optimization—the number of molecules evaluated by the oracle—is rarely discussed, despite being an essential consideration for realistic discovery applications.

To fill this gap, we have created an open-source benchmark for **p**ractical **m**olecular **o**ptimization, `PMO`, to facilitate the transparent and reproducible evaluation of algorithmic advances in molecular optimization. This paper thoroughly investigates the performance of 25 molecular design algorithms on 23 single-objective (scalar) optimization tasks with a particular focus on sample efficiency. Our results show that most "state-of-the-art" methods fail to outperform their predecessors under a limited oracle budget allowing 10K queries and that no existing algorithm can efficiently solve certain molecular optimization problems in this setting. We analyze the influence of the optimization algorithm choices, molecular assembly strategies, and oracle landscapes on the optimization performance to inform future algorithm development and benchmarking. `PMO` provides a standardized experimental setup to comprehensively evaluate and compare new molecule optimization methods with existing ones. All code can be found at `https://github.com/wenhao-gao/mol_opt`.

## 1 Introduction

Designing new functional molecules is a constrained multi-objective optimization problem that aims to find molecules with desired properties such as selective inhibition against a disease target, with additional desiderata and constraints to ensure the structures are stable and synthesizable. The importance of molecular design problems has attracted significant efforts to develop systematical molecular design methodologies instead of exhaustive searches, leveraging combinatorial optimization algorithms [1, 2], predictive machine learning models [3, 4], and generative models [5, 6]. Especially in recent years, we have witnessed significant progress in solving challenging problems across

various aspects of computational molecular optimizations, such as achieving high validity [7, 8, 9], diversity [10], and, most recently, synthesizability [11, 12].

Despite the exciting progress in the field and the abundance of new methods proposed, how these algorithms compare against each other remains unclear. Most method development papers and existing benchmarks such as Guacamol [13], Therapeutics Data Commons (TDC) [14] and Tripp et al.'s [15] suffer from at least one of three problems: (1) Lack of consideration of the oracle budget: Many papers [16, 17, 18] do not report how many times the oracle function is called to achieve the reported results (i.e., how many candidate molecules were evaluated), except in rare cases [19, 20, 21, 22, 23], despite this range spanning orders of magnitude. As most valuable oracles—experiments or high-accuracy simulations—require substantial costs, it is vital to identify the desired compound with as few oracle calls as possible. (2) Trivial oracles: Some papers only report results on trivial oracles [17] like quantitative estimate of drug-likeness (QED) [24][1] or penalized octanol-water partition coefficient (LogP)[2], other papers even introduce new self-designed tasks [18, 21], which obfuscates a comparison to prior work. (3) Randomness: Another complication is that many algorithms are not deterministic and exhibit significant run-to-run variation, so reporting results from several independent trials is essential. All of the existing benchmarks examined no more than five methods due to the significant variation between molecular optimization algorithms. Thus we still lack a unified benchmark to assess which methods are beneficial in a realistic discovery scenario.

This paper presents a new reproducible large-scale experimental study with a sound experimental protocol for molecular design, PMO. We have benchmarked 25 methods across 23 various widely-used oracle functions, with each of them tuned and run for multiple independent trials. To consider a combination of optimization ability and sample efficiency, we limit the number of maximum oracle calls up to 10,000 queries and measure model performance with the area under the curve (AUC) of the top-10 average performance versus oracle calls. Our results show that none of the existing molecular optimization algorithms are efficient enough to solve a *de novo* molecular optimization problem within a realistic oracle budget of hundreds of experiments, and "state-of-the-art" methods often fail to outperform their predecessors. We analyze the algorithmic contribution and the influence of oracle landscapes on optimization performance to inform future algorithm development and benchmarking. Our results highlight the necessity of standardized experimental reporting, including independent replicates and extensive hyperparameter tuning. We envision that the PMO benchmark will make molecular optimizations more accessible and reproducible, thereby facilitating algorithmic advances and, ultimately, the broader adoption of molecular optimization techniques in experimental drug and materials discovery workflows.

## 2   Algorithms

A molecular optimization method has two major components: (1) a molecular assembly strategy that defines the chemical space by assembling a digital representation of compounds, and (2) an optimization algorithm that navigates this chemical space. This section will first introduce common strategies to assemble molecules, then introduce the benchmarked molecular optimization methods based on the core optimization algorithms. Table 1 summarizes current molecular design methods categorized based on assembly strategy and optimization method, including but not limited to the methods included in our baseline. We emphasize that our goal is not to make an exhaustive list but to include a group of methods that are representative enough to obtain meaningful conclusions.

### 2.1   Preliminaries

In this paper, we limit our scope to general-purpose single-objective molecular optimization methods focusing on small organic molecules with scalar properties with some relevance to therapeutic design.

---

[1]As demonstrated in this benchmark later, QED is likely to have a global maximum of 0.948 and even random sampling could reach that value. It is disabled to meaningfully distinguish different algorithms.

[2]LogP is unbounded and the relationship between LogP values and molecular structures is fairly simple: adding carbons monotonically increases the estimated LogP value [25, 20]. This simple strategy makes the performance in LogP highly depend on the chemical space definition and the number of steps allowed, and provides no insights for distinguish algorithms' optimization ability. Besides, simply maximizing LogP is not a meaningful goal in drug design. Therefore, we exclude LogP in this benchmark.

Table 1: Representative molecule generation methods, categorised based on the molecular assembly strategies and the optimization algorithms. Columns are various molecular assembly strategies while rows are different optimization algorithms.

| | SMILES | SELFIES | Graph (atom) | Graph (fragment) | Synthesis |
|---|---|---|---|---|---|
| GA | SMILES-GA [13] | GA+D [17] STONED [29] | - | Graph-GA [1] | SynNet [12] |
| MCTS | - | - | Graph-MCTS [1] | - | - |
| BO | BOSS [30] | - | - | GPBO [15] | ChemBO [19] |
| VAE | SMILES-VAE [6] | SELFIES-VAE [22] | - | JTVAE [8] | DoG-AE [11] |
| GAN | ORGAN [31] | - | MolGAN [32] | - | - |
| SBM | - | - | - | GFlowNet [10] MARS [2] | - |
| HC | SMILES LSTM [13] | SELFIES LSTM | - | MIMOSA [33] | DoG-Gen [11] |
| RL | REINVENT [5] | SELFIES-REINVENT | MolDQN [16] GCPN [25] | RationaleRL [34] FREED [35] | PGFS [18] REACTOR [36] |
| GRAD | - | Pasithea [37] | - | DST [20] | - |

Formally, we can formulate such a molecular design problem as an optimization problem:

$$m^* = \arg \max_{m \in \mathcal{M}} \mathcal{O}(m), \tag{1}$$

where $m$ is a molecular structure, $\mathcal{M}$ denotes the design space called chemical space that comprises all possible candidate molecules. The size of $\mathcal{M}$ is impractically large, e.g., $10^{60}$ [26]. We assume we have access to the ground truth value of a property of interest denoted by $\mathcal{O}(m) : \mathcal{M} \to \mathcal{R}$, where an oracle, $\mathcal{O}$, is a black-box function that evaluates certain chemical or biological properties of a molecule $m$ and returns the ground truth property $\mathcal{O}(m)$ as a scalar. Note that neither the analytic form of oracles nor the derivatives of the properties are accessible. The most practical oracles—experiments or high-accuracy simulations— typically require substantial costs. An algorithm able to optimize the oracle within a reasonable budget is thus necessary for automating the design of molecules to achieve high-level automated chemical design (ACD) [27] or function-oriented autonomous synthesis [28].

## 2.2 Molecular assembly strategies

**String-based**. String-based assembly strategies represent molecules as strings and explore chemical space by modifying strings directly: character-by-character, token-by-token, or through more complex transformations based on a specific grammar. We include two types of string representations: (1) Simplified Molecular-Input Line-Entry System (SMILES) [38], a linear notation describing the molecular structure using short ASCII strings based on a graph traversal algorithm; (2) SELF-referencIng Embedded Strings (SELFIES) [9], which avoids syntactical invalidity by enforcing the chemical validity rules in a formal grammar table.

**Graph-based**. Two-dimensional (2D) graphs can intuitively define molecular identities to a first approximation (ignoring stereochemistry[3]): the nodes and edges represent the atoms and bonds. There are two main assembling strategies for molecular graphs: (1) an atom-based assembly strategy [16] that adds or modifies atoms and bonds one at a time, which covers all valid chemical space; (2) a fragment-based assembling strategy [8] that summarizes common molecular fragments and operates one fragment at a time. Note that fragment-based strategy could also include atom-level operation.

**Synthesis-based**. Most of the above assembly strategies can cover a large chemical space, but an eventual goal of molecular design is to physically test the candidate; thus, a desideratum is to explore synthesizable candidates only. Designing molecules by assembling synthetic pathways from commercially-available starting materials and reliable chemical transformation adds a constraint of synthesizability to the search space. This class can be divided into template-free [11] and template-based [12] based on how to define reliable chemical transformations, but we will not distinguish between them in this paper as synthesis-based strategy is relatively less explored in general.

---

[3]Incorporating certain stereochemical information in 2D molecular graphs is possible through various approaches [39, 40, 41].

## 2.3 Optimization algorithms

**Screening** (a.k.a. virtual screening) involves searching over a pre-enumerated library of molecules. We include Screening as a baseline, which randomly samples ZINC 250k [42]. Model-based screening [43, 44, 45, 46, 3, 47] instead trains a surrogate model and prioritizes molecules that are scored highly by the surrogate to accelerate screening. We adopt the implementation from the original paperof MolPAL [3] and treat it as a model-based version of screening.

**Genetic Algorithm (GA)** is a popular heuristic algorithm inspired by natural evolutionary processes. It combines *mutation* and/or *crossover* perturbing a *mating pool* to enable exploration in the design space. We include SMILES GA [48] that defines actions based on SMILES context-free grammar and a modified version of STONED [29] that directly manipulates tokens in SELFIES strings. Unlike the string-based GAs that only have mutation steps, Graph GA [1] derives crossover rules from graph matching and includes both atom- and fragment-level mutations. Finally, we include SynNet [12] as a synthesis-based example that applies a genetic algorithm on binary fingerprints and decodes to synthetic pathways. We adopt the implementation of SMILES GA and Graph GA from Guacamol [13], STONED, and SynNet from the original paper. We also include the original implementation of a deep learning enhanced version of SELFIES-based GA from [17] and label it as GA+D.

**Monte-Carlo Tree Search (MCTS)** locally and randomly searches each branch of the current state (e.g., a molecule or partial molecule) and selects the most promising ones (those with highest property scores) for the next iteration. Graph MCTS [1] is an MCTS algorithm based on atom-level searching over molecular graphs. We adopt the implementation from Guacamol [13].

**Bayesian optimization (BO)** [49] is a large class of method that builds a surrogate for the objective function using a Bayesian machine learning technique, such as Gaussian process (GP) regression, then uses an *acquisition function* combining the surrogate and uncertainty to decide where to sample, which is naturally model-based. However, as BO usually leverages a non-parametric model, it scales poorly with sample size and feature dimension [50]. We included a string-based model, BO over String Space (BOSS) [30], and a synthesis-based model, ChemBO [19], but do not obtain meaningful results even with early stopping potentially due to the poor scaling of the string subsequence kernel (SSK) (see Section B.3 for early stopping setting, and Section B.33 for more analysis). Finally, we adopt Gaussian process Bayesian optimization (GP BO) [15] that optimizes the GP acquisition function with Graph GA methods in an inner loop. The implementation is from the original paper, and we treat it as a model-based version of Graph GA. Note that we categorize methods that apply BO to optimize molecules in latent space as a separate class below.

**Variational autoencoders (VAEs)** [51] are a class of generative method that maximize a lower bound of the likelihood (evidence lower bound (ELBO)) instead of estimating the likelihood directly. A VAE typically learns to map molecules to and from real space to enable the indirect optimization of molecules by numerically optimizing latent vectors, most commonly with BO [52]. SMILES-VAE [6] uses a VAE to model molecules represented as SMILES strings, and is implemented in MOSES [53]. We adopt the identical architecture to model SELFIES strings and denote it as SELFIES-VAE. JT-VAE [8] abstracts a molecular graph into a junction tree (i.e., a cycle-free structure), and design message passing network as the encoder and tree-RNN as the decoder. DoG-AE [11] uses Wasserstein autoencoder (WAE) to learn the distribution of synthetic pathways. Note that we include a set of vanilla methods for each kind while many variants have emerged, such as [23] and [22]. We leave the validation of variants for the future development of this benchmark.

**Score-based modeling (SBM)** formulates the problem of molecule design as a sampling problem where the target distribution is a function of the target property, featured by Markov-chain Monte Carlo (MCMC) methods that construct Markov chains with the desired distribution as their equilibrium distribution. MARkov molecular Sampling (MARS) [2] is such an example that leverages a graph neural network to propose action steps adaptively in an MCMC with an annealing scheme. Generative Flow Network (GFlowNet) [10] views the generative process as a flow network and trains it with a temporal difference-like loss function based on the conservation of flow. By matching the property of interest with the volume of the flow, generation can sample a distribution proportional to the target distribution.

**Hill climbing (HC)** is an iterative learning method that incorporates the generated high-scored molecules into the training data and fine-tunes the generative model for each iteration. It is a variant

of the cross-entropy method [54], and can also be seen as a variant of REINFORCE [55] with a particular reward shaping. We adopt SMILES-LSTM-HC from Guacamol [13] that leverages a LSTM to learn the molecular distribution represented in SMILES strings, and modifies it to a SELFIES version denoted as SELFIES-LSTM-HC. MultI-constraint MOlecule SAmpling (MIMOSA) [33] leverages a graph neural network to predict the identity of a masked fragment node and trains it with a HC algorithm. DoG-Gen [11] instead learn the distribution of synthetic pathways as Directed Acyclic Graph (DAGs) with an RNN generator.

**Reinforcement Learning (RL)** learns how intelligent agents take actions in an environment to maximize the cumulative reward by transitioning through different states. In molecular design, a state is usually a partially generated molecule; actions are manipulations at the level of graphs or strings; rewards are defined as the generated molecules' property of interest. REINVENT [5] adopts a policy-based RL approach to tune RNNs to generate SMILES strings. We adopt the implementation from the original paper, and modify it to generate SELFIES strings, SELFIES-REINVENT. MolDQN [16] uses a deep Q-network to generate molecular graph in an atom-wise manner.

**Gradient ascent (GRAD)** methods learn to estimate the gradient direction based on the landscape of the molecular property over the chemical space, and back-propagate to optimize the molecules. Pasithea [37] exploits an MLP to predict properties from SELFIES strings, and back-propagate to modify tokens. Differentiable scaffolding tree (DST) [20] abstracts molecular graphs to scaffolding trees and leverages a graph neural network to estimate the gradient. We adopted the implementation from the original papers and modify them to update the surrogates online as data are acquired.

## 3 Experiments

### 3.1 Benchmark setup

This section introduces the setup of `PMO` benchmark. The main idea behind `PMO` is the pursuit of an ideal *de novo* molecular optimization algorithm that exhibits strong optimization ability, sample efficiency, generalizability to various optimization objectives, and robustness to hyperparameter selection and random seeds.

**Oracle**: To examine the generalizability of methods, we aim to include a broad range of pharmaceutically-relevant oracle functions. Systematic categorization of oracles based on their landscape is still challenging due to the complicated relationship between molecular structure and function. We have included the most commonly used oracles (see a recent discussion of commonly-used oracles in [56]). Several have been described as "trivial", but we assert this is only true when the number of oracle queries is not controlled. In total, `PMO` includes 23 oracle functions: QED [24], DRD2 [5], GSK3$\beta$, JNK3 [57], and 19 oracles from Guacamol [13]. QED is a relatively simple heuristic function that estimates if a molecule is likely to be a drug based on if it contains some "red flags". DRD2, GSK3$\beta$, and JNK3 are machine learning models (support vector machine (SVM), random forest (RF)) fit to experimental data to predict the bioactivities against their corresponding disease targets. Guacamol oracles are designed to mimic the drug discovery objectives based on multiple considerations, called multi-property objective (MPO), including similarity to target molecules, molecular weights, CLogP, etc. All oracle scores are normalized from 0 to 1, where 1 is optimal. Recently, docking scores that estimate the binding affinity between ligands and proteins have been adopted as oracles [58, 14, 59]. However, as the simulations are more costly than above ones but are still coarse estimates that do not reflect true bioactivity, we leave it to future work.

**Metrics**: To consider the optimization ability and sample efficiency simultaneously, we report the area under the curve (AUC) of top-$K$ average property value versus the number of oracle calls (*AUC top-$K$*) as the primary metric to measure the performance. Unlike using top-$K$ average property, AUC rewards methods that reach high values with fewer oracle calls. We use $K = 10$ in this paper as it is useful to identify a small number of distinct molecular candidates to progress to later stages of development. We limit the number of oracle calls to 10000, though we expect methods to optimize well within hundreds of calls when using experimental evaluations. The reported values of AUCs are min-max scaled to $[0, 1]$.

**Data**: We restrict all our methods to using the ZINC 250K dataset only whenever a database is required, which contains around 250K molecules sampled from the ZINC database [42] for its pharmaceutical relevance, moderate size, and popularity. Screening and MolPAL search over this

Table 2: Performance of ten best performing molecular optimization methods based on mean AUC Top-10. We report the mean and standard deviation of **AUC Top-10** from 5 independent runs. The best model in each task is labeled bold. Full results are in the Appendix A.

| Method
Assembly | REINVENT
SMILES | Graph GA
Fragments | REINVENT
SELFIES | GP BO
Fragments | STONED
SELFIES |
|---|---|---|---|---|---|
| albuterol_similarity | 0.882± 0.006 | 0.838± 0.016 | 0.826± 0.030 | **0.898± 0.014** | 0.745± 0.076 |
| amlodipine_mpo | 0.635± 0.035 | **0.661± 0.020** | 0.607± 0.014 | 0.583± 0.044 | 0.608± 0.046 |
| celecoxib_rediscovery | 0.713± 0.067 | 0.630± 0.097 | 0.573± 0.043 | **0.723± 0.053** | 0.382± 0.041 |
| deco_hop | 0.666± 0.044 | 0.619± 0.004 | 0.631± 0.012 | 0.629± 0.018 | 0.611± 0.008 |
| drd2 | 0.945± 0.007 | 0.964± 0.012 | 0.943± 0.005 | 0.923± 0.017 | 0.913± 0.020 |
| fexofenadine_mpo | 0.784± 0.006 | 0.760± 0.011 | 0.741± 0.002 | 0.722± 0.005 | **0.797± 0.016** |
| gsk3b | **0.865± 0.043** | 0.788± 0.070 | 0.780± 0.037 | 0.851± 0.041 | 0.668± 0.049 |
| isomers_c7h8n2o2 | 0.852± 0.036 | 0.862± 0.065 | 0.849± 0.034 | 0.680± 0.117 | 0.899± 0.011 |
| isomers_c9h10n2o2pf2cl | 0.642± 0.054 | 0.719± 0.047 | 0.733± 0.029 | 0.469± 0.180 | 0.805± 0.031 |
| jnk3 | **0.783± 0.023** | 0.553± 0.136 | 0.631± 0.064 | 0.564± 0.155 | 0.523± 0.092 |
| median1 | **0.356± 0.009** | 0.294± 0.021 | 0.355± 0.011 | 0.301± 0.014 | 0.266± 0.016 |
| median2 | 0.276± 0.008 | 0.273± 0.009 | 0.255± 0.005 | **0.297± 0.009** | 0.245± 0.032 |
| mestranol_similarity | 0.618± 0.048 | 0.579± 0.022 | 0.620± 0.029 | **0.627± 0.089** | 0.609± 0.101 |
| osimertinib_mpo | **0.837± 0.009** | 0.831± 0.005 | 0.820± 0.003 | 0.787± 0.006 | 0.822± 0.012 |
| perindopril_mpo | 0.537± 0.016 | 0.538± 0.009 | 0.517± 0.021 | 0.493± 0.011 | 0.488± 0.011 |
| qed | 0.941± 0.000 | 0.940± 0.000 | 0.940± 0.000 | 0.937± 0.000 | 0.941± 0.000 |
| ranolazine_mpo | 0.760± 0.009 | 0.728± 0.012 | 0.748± 0.018 | 0.735± 0.013 | **0.765± 0.029** |
| scaffold_hop | **0.560± 0.019** | 0.517± 0.010 | 0.525± 0.013 | 0.548± 0.019 | 0.521± 0.034 |
| sitagliptin_mpo | 0.021± 0.003 | **0.433± 0.075** | 0.194± 0.121 | 0.186± 0.055 | 0.393± 0.083 |
| thiothixene_rediscovery | 0.534± 0.013 | 0.479± 0.025 | 0.495± 0.040 | **0.559± 0.027** | 0.367± 0.027 |
| troglitazone_rediscovery | **0.441± 0.032** | 0.390± 0.016 | 0.348± 0.012 | 0.410± 0.015 | 0.320± 0.018 |
| valsartan_smarts | **0.178± 0.358** | 0.000± 0.000 | 0.000± 0.000 | 0.000± 0.000 | 0.000± 0.000 |
| zaleplon_mpo | **0.358± 0.062** | 0.346± 0.032 | 0.333± 0.026 | 0.221± 0.072 | 0.325± 0.027 |
| Sum | 14.196 | 13.751 | 13.471 | 13.156 | 13.024 |
| Rank | 1 | 2 | 3 | 4 | 5 |
| Method
Assembly | LSTM HC
SMILES | SMILES GA
SMILES | SynNet
Synthesis | DoG-Gen
Synthesis | DST
Fragments |
| albuterol_similarity | 0.719± 0.018 | 0.661± 0.066 | 0.584± 0.039 | 0.676± 0.013 | 0.619± 0.020 |
| amlodipine_mpo | 0.593± 0.016 | 0.549± 0.009 | 0.565± 0.007 | 0.536± 0.003 | 0.516± 0.007 |
| celecoxib_rediscovery | 0.539± 0.018 | 0.344± 0.027 | 0.441± 0.027 | 0.464± 0.009 | 0.380± 0.006 |
| deco_hop | **0.826± 0.017** | 0.611± 0.006 | 0.613± 0.009 | 0.800± 0.007 | 0.608± 0.008 |
| drd2 | 0.919± 0.015 | 0.908± 0.019 | **0.969± 0.004** | 0.948± 0.001 | 0.820± 0.014 |
| fexofenadine_mpo | 0.725± 0.003 | 0.721± 0.015 | 0.761± 0.015 | 0.695± 0.003 | 0.725± 0.005 |
| gsk3b | 0.839± 0.015 | 0.629± 0.044 | 0.789± 0.032 | 0.831± 0.021 | 0.671± 0.032 |
| isomers_c7h8n2o2 | 0.485± 0.045 | **0.913± 0.021** | 0.455± 0.031 | 0.465± 0.018 | 0.548± 0.069 |
| isomers_c9h10n2o2pf2cl | 0.342± 0.027 | **0.860± 0.065** | 0.241± 0.064 | 0.199± 0.016 | 0.458± 0.063 |
| jnk3 | 0.661± 0.039 | 0.316± 0.022 | 0.630± 0.034 | 0.595± 0.023 | 0.556± 0.057 |
| median1 | 0.255± 0.010 | 0.192± 0.012 | 0.218± 0.008 | 0.217± 0.001 | 0.232± 0.009 |
| median2 | 0.248± 0.008 | 0.198± 0.005 | 0.235± 0.006 | 0.212± 0.000 | 0.185± 0.020 |
| mestranol_similarity | 0.526± 0.032 | 0.469± 0.029 | 0.399± 0.021 | 0.437± 0.007 | 0.450± 0.027 |
| osimertinib_mpo | 0.796± 0.002 | 0.817± 0.011 | 0.796± 0.003 | 0.774± 0.002 | 0.785± 0.004 |
| perindopril_mpo | 0.489± 0.007 | 0.447± 0.013 | **0.557± 0.011** | 0.474± 0.002 | 0.462± 0.008 |
| qed | 0.939± 0.000 | 0.940± 0.000 | **0.941± 0.000** | 0.934± 0.000 | 0.938± 0.000 |
| ranolazine_mpo | 0.714± 0.008 | 0.699± 0.026 | 0.741± 0.010 | 0.711± 0.006 | 0.632± 0.054 |
| scaffold_hop | 0.533± 0.012 | 0.494± 0.011 | 0.502± 0.012 | 0.515± 0.005 | 0.497± 0.004 |
| sitagliptin_mpo | 0.066± 0.019 | 0.363± 0.057 | 0.025± 0.014 | 0.048± 0.008 | 0.075± 0.032 |
| thiothixene_rediscovery | 0.438± 0.008 | 0.315± 0.017 | 0.401± 0.019 | 0.375± 0.004 | 0.366± 0.006 |
| troglitazone_rediscovery | 0.354± 0.016 | 0.263± 0.024 | 0.283± 0.008 | 0.416± 0.019 | 0.279± 0.019 |
| valsartan_smarts | 0.000± 0.000 | 0.000± 0.000 | 0.000± 0.000 | 0.000± 0.000 | 0.000± 0.000 |
| zaleplon_mpo | 0.206± 0.006 | 0.334± 0.041 | 0.341± 0.011 | 0.123± 0.016 | 0.176± 0.045 |
| Sum | 12.223 | 12.054 | 11.498 | 11.456 | 10.989 |
| Rank | 6 | 7 | 8 | 9 | 10 |

database; generative models such as VAEs, LSTMs are pretrained on this database; fragments required for JT-VAE, MIMOSA, DST are extracted from this database.

**Other details**: We tuned hyperparameters for most methods on the average AUC Top-10 from 3 independent runs of two Guacamol tasks: zaleplon_mpo and perindopril_mpo. Reported results are from 5 independent runs with various random seeds. All data, oracle functions, and metric evaluations are taken from the Therapeutic Data Commons (TDC) [14] (https://tdcommons.ai) and more details are described in Appendix. Note that the implementation of sitagliptin_mpo and zaleplon_mpo are different from the ones in Guacamol [13].

Table 3: The ranking of each methods based on different metrics.

| Method | AUC Top-1 | AUC Top-10 | AUC Top-100 | Top-1 | Top-10 | Top-100 | Mean |
|---|---|---|---|---|---|---|---|
| REINVENT | 1 | 1 | 1 | 1 | 1 | 1 | 1 |
| Graph GA | 2 | 2 | 2 | 3 | 2 | 3 | 2.33 |
| SELFIES-REINVENT | 3 | 3 | 4 | 4 | 3 | 2 | 3.16 |
| SMILES-LSTM-HC | 5 | 6 | 7 | 2 | 4 | 4 | 4.66 |
| GP BO | 4 | 4 | 5 | 6 | 5 | 5 | 4.83 |
| STONED | 6 | 5 | 3 | 7 | 7 | 6 | 5.66 |
| DoG-GEN | 7 | 9 | 11 | 5 | 6 | 7 | 7.5 |
| SMILES GA | 9 | 7 | 6 | 10 | 8 | 8 | 8 |
| DST | 11 | 10 | 9 | 9 | 10 | 9 | 9.66 |
| SynNet | 8 | 8 | 8 | 11 | 11 | 14 | 10 |
| SELFIES-LSTM-HC | 13 | 14 | 13 | 8 | 9 | 11 | 11.33 |
| MIMOSA | 14 | 12 | 10 | 14 | 12 | 10 | 12 |
| MARS | 12 | 11 | 12 | 12 | 13 | 13 | 12.16 |
| MolPAL | 10 | 13 | 15 | 13 | 15 | 16 | 13.66 |
| GA+D | 23 | 17 | 14 | 15 | 14 | 12 | 15.83 |
| DoG-AE | 15 | 15 | 17 | 17 | 17 | 17 | 16.33 |
| GFlowNet | 20 | 16 | 16 | 19 | 16 | 15 | 17 |
| SELFIES-VAE | 16 | 18 | 21 | 16 | 18 | 21 | 18.33 |
| Screening | 17 | 19 | 19 | 18 | 19 | 19 | 18.5 |
| SMILES-VAE | 18 | 20 | 20 | 20 | 20 | 20 | 19.66 |
| GFlowNet-AL | 22 | 22 | 18 | 23 | 21 | 18 | 20.66 |
| Pasithea | 19 | 21 | 23 | 21 | 22 | 22 | 21.33 |
| JT-VAE | 21 | 23 | 22 | 22 | 23 | 23 | 22.33 |
| Graph MCTS | 24 | 24 | 24 | 24 | 24 | 24 | 24 |
| MolDQN | 25 | 25 | 25 | 25 | 25 | 25 | 25 |

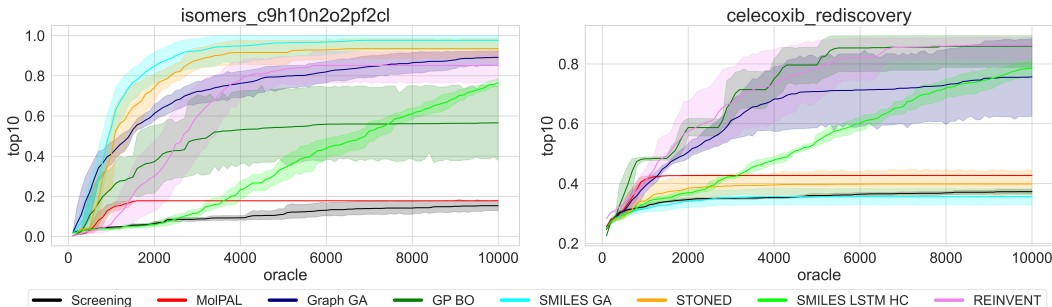

Figure 1: The optimization curves of top-10 average on optimizing isomer_c9h10n2o2pf2cl and celecoxib_rediscovery, as the representation of isomer-type and similarity-type oracles. Only 8 methods are displayed for clarity and full results are in the Appendix A.

## 3.2 Results & Analysis

The primary results are summarized in Table 2 and 3. For clarity, we only show the ten best-performing models in the table. We show a selective set of optimization curves in Figure 1. The remaining results are in the Appendix A and D.

**Sample efficiency matters.** A first observation from the results is that none of the methods we implemented can optimize the simple toy objectives within hundreds of oracle calls under our experimental settings, except some trivial ones like QED, DRD2, and osimertinib_mpo, which emphasize the need for more efficient molecular optimization algorithms. By comparing the ranking of AUC Top-10 and Top-10, we notice some methods have significantly different relative performances. For example, SMILES LSTM HC, which used to be seen as comparable to Graph GA, actually requires more oracle queries to achieve the same level of performance, while a related algorithm, REINVENT, requires far fewer (see Figure 1). These differences indicate the training algorithm of REINVENT is more efficient than HC, emphasizing the importance of AUC Top-10 as an evaluation metric. In addition, methods that assemble molecules either token-by-token or atom-by-atom from a single start point, such as GA+D, MolDQN, and Graph MCTS, are most data-inefficient. Those methods potentially cover broader chemical space and include many undesired candidates, such as unstable or unsynthesizable ones, which wastes a significant portion of the oracle budget and also imposes a strong requirement on the oracles' quality.

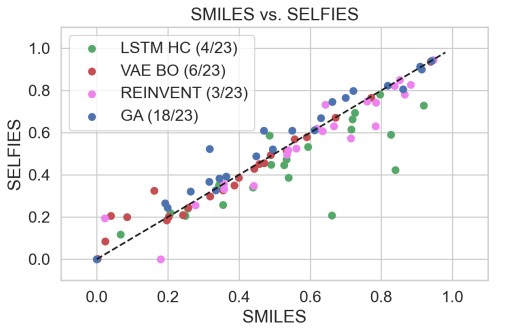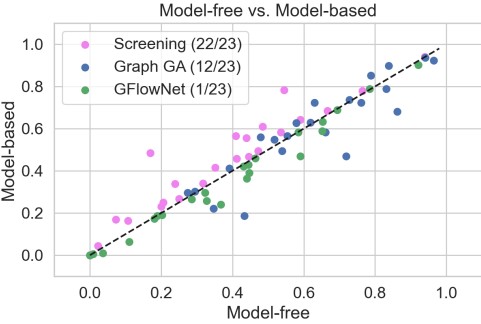

(a) Comparison between SMILES- and SELFIES-based methods. Note GA is not a head-to-head comparison.
(b) Comparison between model-free and corresponding model-based methods.

Figure 2: Each point represents the AUC Top-10 of one task, with x-axis the SMILES variant and y-axis the SELFIES variant of the same method. Colors are labeled by the optimization algorithms. The fractions of the tasks above the parity line are in parentheses.

**Older algorithms are still powerful.** As shown in Table 2 and 3, the best-performing algorithms are REINVENT and Graph GA among all the compared methods, despite both of them being released several years ago. However, we rarely see model development papers list these two methods as baselines. The absence of a thorough benchmark has obfuscated the fact that newer models published in top AI conferences do not seem to offer an improvement in performance by our metrics. Of course, we should acknowledge that some of the methods are developed to solve other problems in molecular optimization, such as strings' validity or synthesizability, and some might have opened new avenues to tackle the problem that could potentially be more efficient when mature. Still, some of the field's efforts and resources might be wasted due to a lack of a thorough and standardized benchmark.

**There are no obvious shortcomings of SMILES.** SELFIES was designed as a substitute of SMILES to solve the syntactical invalidity problem met in SMILES representation and has been adopted by a number of recent studies. However, our head-to-head comparison of string-based methods, especially the ones leveraging language models, shows that most SELFIES variants cannot outperform their corresponding SMILES-based methods in terms of optimization ability and sample efficiency (Figure 2a). We do observe some early methods like the initial version of SMILES VAE [6] (2016) and ORGAN [31] (2017) struggle to propose valid SMILES strings, but this is not an issue for more recent methods. We believe this is partially because current language models are better able to learn the grammar of SMILES strings, which has flattened the advantage of SELFIES. Further, as shown in Appendix D.1, more combinations of SELFIES tokens don't necessarily explore larger chemical space but might map to a small number of valid molecules that can be represented by truncated SELFIES strings, which implies that there are still syntax requirements in generating SELFIES strings to achieve effective exploration.

On the other hand, we observe a clear advantage of SELFIES-based GA compared to SMILES-based one, which indicates that SELFIES has an advantage over SMILES when we need to design the rules to manipulate the sequence. However, we should note that the comparison is not head-to-head, as GAs' performances highly depend on the mutation and crossover rule design, but not the representation. Graph GA's mutation rules are also encoded in SMARTS strings and operate on SMILES strings, which can also be seen as SMILES modification steps. Overall, when we need to design the generative action manually, the assembly strategy that could derive desired transformation more intuitively should be preferred.

**Model-based methods are potentially more efficient but need careful design.** It is widely recognized in the RL community that model-based optimization methods that explicitly leverage a predictive model ("world model") are more sample efficient than the model-free ones [60]. Our results on MolPAL and screening verify the principle that training a predictive model is beneficial compared to random sampling (see Figure 2b). However, the results of Graph GA (model-based variant: GP BO) and GFlowNet (model-based variant: GFlowNet-AL) indicate that simply adding a predictive model might not necessarily be helpful. GP BO outperformed Graph GA in 12 tasks among 23, but Graph GA outperformed GP BO in the summation. GFlowNet outperformed GFlowNet-AL in almost every task. From the step-wise increment behavior (see Figure 1) and hyper-parameter

Figure 3: The heatmap and the clustering of oracles based on relative AUC Top-10. Relative AUC Top-10 is computed by normalizing AUC Top-10 values to a range from the lowest and the highest value within the task. The zaleplon_mpo and sitagliptin_mpo are multi-objective versions of isomer functions [13], while all other MPOs are based on similarity. Clear patterns emerge between a large cluster of similarity-based oracles, four isomer-based oracles, and other non-clustered ones. Different types of landscape are more suitable for different kinds of methods to explore. The cluster tree was calculated with unweighted pair group method with arithmetic mean (UPGMA) using Euclidean distance.

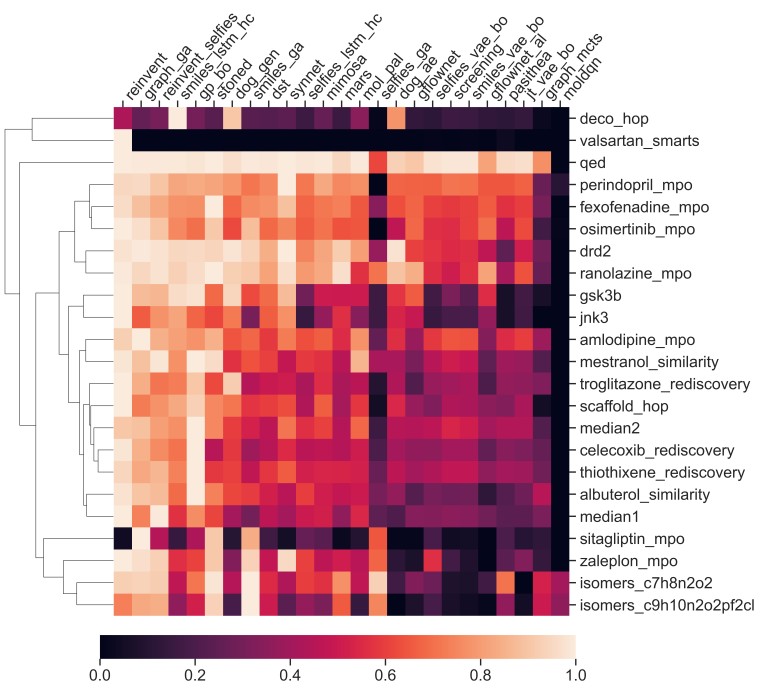

tuning of GP BO (Appendix D.2), we conclude that the performance bottleneck is mainly the quality of the predictive model. Further, GFlowNet-AL adopts a relatively naive model-based strategy that may suppress exploitation, especially when the model is not well-trained. Overall, we observe that model-based optimization algorithms have the potential to be more sample efficient but require careful design of the inner- and outer-loop optimization algorithms so the model does not lead the search astray.

**Different types of methods are more suitable for different kinds of landscapes.** As shown from Figure 3 and Table 2, we find that there are some clear clusters of oracles based on the relative performance of methods. One clear pattern is that string-based GAs, such as SMILES GA and STONED, reach superior relative performance in tasks involving isomer functions, including isomer_c7h8n2o2, isomer_c9h10n2o2pf2cl, sitagliptin_mpo, and zaleplon_mpo. Isomer-type oracles are summations of atomic contribution, while all other MPOs are mainly based on similarity measured by fingerprints, and they generally have closer relative performance. Among similarity-based oracles, the ones including logP and TPSA, such as fexofenadine_mpo and osimertinib_mpo, are clustered together against more naive similarities such as the rediscovery and median ones. The machine learning oracles predicting bioactivities belong to the same cluster of similarity-based oracles. While QED is too trivial that almost all methods reach very close values, deco_hop, valsartan_smarts, scaffold_hop that are designed based on whether a molecule contains a substructure have varied performance. The results suggest that different types of landscape are better explored by different kinds of methods, such as string-based GA on isomer-type oracles. It is not evident which type of oracle is closest to a "true" pharmaceutical design objective, which is likely more complex and challenging to optimize; we leave further investigation on oracle landscapes and their influence on optimization to future work.

**Hyperparameter reoptimization and multiple runs are required when reporting results.** We also observed that the optimal set of hyper-parameters is always not the default ones suggested by a method's original paper (see Appendix D.2). For example, REINVENT's performance is highly dependent on $\sigma$; we found the best-performing value to be much larger than the values suggested in the original paper (see Figure 15 and 14) [5]. We conclude that this is due to unique demands of our setting of limited oracle budget, which was not a goal of the original study, and thus suggest reoptimizing the hyper-parameters whenever the testing environment is changed. Another challenge is the non-determinism of most algorithms. For example, Graph GA suffers from a relatively

large variance due to its random-walk-like exploration, as does GP BO. If the oracle were a costly experimental evaluation, we might consider the worst-case performance as an endpoint to reduce the risk rather than the average performance, highlighting the importance of running multiple independent runs and reporting the distribution of outcomes.

## 4    Conclusions

This paper proposes PMO: a standardized molecular design benchmark focusing on sample efficiency as a key impediment to experimental adoption. We conduct a thorough investigation across 25 methods and 23 objectives to determine the current state-of-the-art, investigate problems, and draw insights for future studies. Our primary observations are that (1) methods considered to be strong baselines, like LSTM HC, may be inefficient in data usage; (2) several older methods, like REINVENT and Graph GA, outperform more recent ones; (3) SELFIES does not seem to offer an immediate benefit in optimization performance compared to SMILES except in GA; (4) model-based methods have the potential to be more sample efficient but require careful design of the inner-loop, outer-loop, and the predictive model; and (5) different optimization algorithms may excel at different tasks, determined by the landscapes of oracle functions; which algorithm to select is still dependent on the use case and the type of tasks.

We acknowledge several limitations of the current study: we cannot exhaustively explore every method and thoroughly tune every hyperparameter, the representative methods we implement might not be the best-in-class among all possible variants, our conclusion might be biased toward similarity-based oracles, and we are not thoroughly investigating other important quantities such as synthesizability [61] and diversity [14]. We also emphasize that our experiments consider the number of oracle calls from scratch, i.e., the data used to train the surrogate models in model-based methods are counted in the total budget. If a dataset has been collected previously, it may be prudent to train a surrogate model on this information and use a model-based method as illustrated by Tripp et al. [15]. We will support the continued development of this benchmark to minimize the wasted effort caused by non-reproducibility and poor baselines to boost the field's growth toward solving practical molecular design problems.

We would like to conclude with recommendations for subsequent studies: (1) When comparing baselines, it is important to run algorithms under the same oracle budgets; (2) For general-purpose molecular design algorithms, one should test on multiple types of oracles; (3) Conducting multiple independent runs and reporting the distribution of outcomes is critical for non-deterministic methods; (4) Whenever the tasks and testing environment are changed, hyperparameter tuning is necessary.

## Acknowledgments and Disclosure of Funding

This research was supported by the Office of Naval Research under grant number N00014-21-1-2195 and the Machine Learning for Pharmaceutical Discovery and Synthesis consortium. Any opinions, findings, and conclusions or recommendations expressed in this material are those of the author(s) and do not necessarily reflect the views of the Office of Naval Research. W.G. received additional funding from MIT-Takeda fellowship. T.F. and J.S. were supported by NSF award SCH-2205289, SCH-2014438, IIS-1838042, NIH award R01 1R01NS107291-01. We thank Samuel Goldman and John Bradshaw for commenting on the manuscript.

## Reproducibility Statement

All code, parameters, and releasable data can be found at `https://github.com/wenhao-gao/mol_opt`, including instructions in a README file. All results generated in this experiment can be found at `https://figshare.com/articles/dataset/Results_for_practival_molecular_optimization_PMO_benchmark/20123453`. Appendix B describe the experimental setup, implementation details, datasets used, and hardware configuration.

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
