# OpenReview forum: "Sample Efficiency Matters: A Benchmark for Practical Molecular Optimization"
_NeurIPS.cc/2022/Track/Datasets_and_Benchmarks — NeurIPS 2022 Datasets and Benchmarks _

### Official Review · Reviewer_J6Z5 · 2022-07-20
**An awesome molecular optimization benchmark providing a chance to evaluate the sample efficiency**

**Rating:** 8
**Confidence:** 4
**Clarity:** This paper is clear and well-organized.

**Strengths:**

(+) This work provides an excellent and convenient toolkit for unified evaluation of molecular optimization algorithms, which is very helpful for new comers in this research area and easy to fairly compare different optimization methods.
(+) This work emphasizes the importance of sample efficiency, which is actually critical in practical molecular optimization applications but has been long ignored in previous molecular optimization method. This insightful point makes a significant contribution in leading researchers in this area to consider more about the real-world application scenarios when developping their methods.
(+) The experimental results show that some old methods are in fact comparable or even more powerful than most recent methods. This discovery is really enlightening and can motivate researchers to rethink the values of old methods.

**Weaknesses:**

(-) This work focuses on the single-objective molecular optimization problem. However, in some real-world applications, we need to solve the multi-objective molecular optimization problem. I hope in the future, the authors can improve the current benchmark by including multi-objective optimization methods and evaluation.
(-) A typo exists in line 76: "demotes" should be "denotes".

**Additional Feedback:**

N/A

**Correctness:**

The claims of this work are sound and well supported by solid experimental results. All evaluation methods and experiment design are reasonable and performed correctly.

**Documentation:**

There are sufficient details and information to reproduce the experimental results.

**Ethics:**

No ethical review is needed.

**Relation To Prior Work:**

This paper clearly discusses the relation to prior work and clarifies the major difference in evaluation metrics between the proposed benchmark and prior work.

**Summary And Contributions:**

This paper creates an open and unified benchmark to fairly evaluate existing molecular optimization algorithms. Different from previously used evaluation metrics, the proposed benchmark particularly focuses on sample efficiency. Comprehensive experimental studies are conducted to evaluate and compare previous molecular optimization methods.

---

> ### Author Response · Authors · 2022-08-15
> **Response to comments from reviewer J6Z5**
>
> Thank you for your comments. Please find our response below. We have uploaded more results in the Appendix and highlighted the important changes in the draft in red.
>
> **Q1**: This work focuses on the single-objective molecular optimization problem. However, in some real-world applications, we need to solve the multi-objective molecular optimization problem. I hope in the future, the authors can improve the current benchmark by including multi-objective optimization methods and evaluation.
>
> **A**: Yes, we agree. We added a sentence to acknowledge the current results are about single scalar objectives in the abstract, and we plan to include multi-objective optimization in the code-base in the future. Also see our comment to Reviewer 5BVJ that many/most “multi-objective” optimization demonstrations have relied on scalarization, making the problem formulation identical to single-objective optimization; we feel that the field has not produced algorithms tailored to true multi-objective optimization.
>
> **Q2**: A typo exists in line 76: "demotes" should be "denotes".
>
> **A**: We have fixed it, thanks for pointing it out.

---

### Official Review · Reviewer_VZwE · 2022-07-25
**A good benchmark for molecular optimization/generation and also some good points are validated by experiments, but some new and useful oracle functions are needed.**

**Rating:** 6
**Confidence:** 4
**Clarity:** This paper is well written and organi…

**Strengths:**

1. Provided a standardized and thorough benchmark for molecular optimization, including 23 already used oracle functions and the introduced optimization ability and sample efficiency in this work.
2. Very thorough summary and comparison of molecular generation methods, even some older algorithms that not very often compared by current algorithms. Therefore, the current benchmark and metrics are important.
3. Suggest firstly the AUC Top-X as a metric instead of current Top-X to consider sample efficiency and optimization ability for PMO task.
4. It's good to compare SMILES with SELFIES using different molecular generation methods and give us a sound conclusion about the two common molecular representation methods.

**Weaknesses:**

Compared with previous benchmark Guacamol, only added QED, DRD2, GSK3, JNK3 and two more functions, however, more relative to real drug discovery functions or properties are still missed. For example, docking scores as they mentioned, synthetic feasibility assessment, and some key ADMET properties. Because most of the current oracle functions are not very useful for real drug discovery, even the ones used in Table 2 of this work.


**Additional Feedback:**

No

**Correctness:**

The evaluation methods and experiment design appropriate and performed correctly as a benchmark.

**Documentation:**

The documentation seems sufficient.

**Ethics:**

I have not found any ethical concerns.

**Relation To Prior Work:**

The comparison with previous methods, such as Guacamol and TDC, are clearly discussed.

**Summary And Contributions:**

The paper provided a standardized and thorough benchmark for molecular optimization. It is good to have such a benchmark for future molecular optimization task to follow and compare. This work benchmarked thoroughly 25 representative generative methods over 23 oracle functions. Compared to previous benchmarks, this work provides several new oracle functions. The author also suggested the AUC Top-X as a metric instead of current Top-X to consider sample efficiency and optimization ability. Moreover, the paper also gave several useful experimental results. The main weakness for this paper is the newly added oracle functions and previous ones are not very useful for real drug discovery MPO (multi-properties optimization) or PMO tasks. I suggest to add docking score, synthetic feasibility assessment, ADMET properties and so on in future work.

---

> ### Author Response · Authors · 2022-08-15
> **Response to comments from reviewer VZwE**
>
> Thank you for your comments. Please find our response below. We have uploaded more results in the Appendix and highlighted the important changes in the draft in red.
>
> **Q1**: New oracles needed.
>
> **A**: We agree that the field generally needs better oracle functions, but the current oracles has proven to be enough to differentiate algorithms’ optimization ability. Note the focus of this paper is that existing algorithms cannot even optimize those toy oracles within a reasonable number of oracle calls. We have explained the choice of our oracles in section 3.1. Docking scores are more costly but still pretty coarse (see [1] for an example). Simply optimizing synthetic feasibility is not itself a useful task, and building/training good oracle functions for synthetic feasibility and key ADMET properties is a challenging machine learning task by itself. Besides, our code base is extensible to any oracle function in TDC’s format [2], thus one can incorporate other potential oracles in the future.
>
> Reference:
> ===
>
> [1] Alon, Assaf, et al. "Structures of the σ2 receptor enable docking for bioactive ligand discovery." Nature 600.7890 (2021): 759-764.
>
> [2] https://tdcommons.ai/functions/oracles/

---

### Official Review · Reviewer_i48g · 2022-07-26
**Paper with insightful experiments but proposed benchmark needs more work**

**Rating:** 4
**Confidence:** 5

**Strengths:**

- *Correct insight about sample efficiency*: many alleged "SOTA" algorithms require so many oracle calls that they would never be used in practice. Pointing this out and trying to correct it is worthwhile.
- *Addresses need for benchmarks in molecule optimization*: I agree with the authors that there are little/no standardized benchmarks which makes comparison difficult, and think that PMO contributes to this (although the difference between it and TDC/Guacamol is not very large).
- *Good metrics*: I like the AUC metrics proposed by the authors, even though I think they have some issues (discussed below). In particular, including full plots of the optimization trajectory is very useful.
- *Extensive experiments*: the authors tested a large number of baseline methods which is good (even though I believe that some methods were not implemented optimally; see below). Quality aside, including a large number of baselines is better than including a smaller number.
- *Good writing*: paper is very clear and well-written

**Weaknesses:**

- Metrics don't include diversity which I think may not make "topk" very meaningful (see correctness)
- Potential problem with setup: budget of 10k seems somewhat arbitrary (see correctness)
- Does not address effect of datasets on performance (see correctness)
- The selection of objective functions is not very diverse, and inherits some strong underlying biases of Guacamol. I worry that PMO functions are not very representative of functions in drug design more generally. (see correctness)
- Some baselines seem to be sub-optimally implemented which I believe results in their performance being understated (see "correctness"). This may sound like a minor issue, but I believe this renders the performance numbers in Tables 2, 3, etc not very meaningful, and therefore will be artificially easy to beat if subsequent papers use PMO to evaluate their algorithms (i.e. the main purpose of baseline algorithms).

**Additional Feedback:**

**Alternative focus of the paper**: I think that defining a good benchmark for molecular optimization is very difficult, so I understand that many critiques made in my review might not be easy to address. If this is the case, one thing you could do is re-package this work as an empirical paper for a cheminformatics journal. I think that a lot of people in chemistry tend to use methods "out of the box" and with a small sample budget. Your results in these settings (namely that GAs/REINVENT are strong choices) would be of great interest to many practitioners, leading to an impactful paper.

**Recommendation for baselines**: I think for this kind of paper it would be much better to have 5 really well-tuned baselines (perhaps a representative method from many classes of algorithms) rather than a large number of moderately-tuned baselines, just as your current set of 23 moderately-tuned baselines would be better than 100 very poorly configured baselines.

**Clarity:**

In general this paper was very well-written so I don't have much feedback. One thing that I think would be helpful though is an equation defining AUC precisely: I think I know what it means but would like to see an equation to ensure that I am interpreting it correctly.

**Correctness:**

Here I will comment on the weaknesses mentioned above, which in general I feel fall under the domain of "correctness".
In general, these comments come from my experience both with many of the baseline algorithms tested here,
and with the objective functions studied; I have spent over 2 years working on molecular optimization and am very familiar with the methods in this paper.

**Metrics**

AUC is a sensible metric to me because it rewards both good final performance and good initial performance.
My concern is about the choice of top-10:
Guacamol + fingerprint-based objective functions in general tend not to be very sensitive to small changes in molecular structure;
in my experience if one high-scoring molecule is known then it is easy to produce many high-scoring variations of that molecule (although some algorithms do this better than others).
This raises 2 concerns:
1. I don't think that measuring an algorithm's ability to produce variations of the same molecule is as important as finding a good initial molecule; variations of this molecule can be produced later. This matches the existing pipeline of "hit identification" and "lead optimization" in real-world drug discovery.
2. Practically, I worry that measuring the "top 10" performance will not be much different than measuring "top 1" performance.

I think that it would be better to mainly report top1 performance: this mitigates any potential concerns about diversity (which you deem "future work" anyway), and would make PMO more closely aligned with the hit identification task in drug discovery.
However, you could instead add some sort of diversity penalty, either as a hard constraint (often hard to incorporate) or a soft penalty (forcing you to pick an arbitrary trade-off between objective value and diversity).
I think that both of these options are better than using the top 10 metric and disregarding diversity.

**Benchmark setup: 10k budget**

I would like to hear the authors' reasons for choosing 10k as the oracle budget. This choice seemed somewhat arbitrary to me, and was not justified in the paper. I personally think that it is quite an awkward number: too high for real-world experiments (where even 1000 is a very high number of sequential experiments), but too low for in-silico screening (where even moderately expensive functions could be evaluated at least $10^5$ times). This makes me worried that the PMO setup does not represent real-world applications very well.

**Benchmark setup: choice of dataset**

To my knowledge, every molecular optimization algorithm uses data in some ways: for example to train a predictive or generative model, but at the very least to pick a "starting point" for iterative algorithms.
In general, it should be clear that data available impacts the difficulty of the optimization problem; problems are easier in general if there is data close to the desired optima, and more difficult if data is far from it.
For example, algorithms like MCMC or genetic algorithms which perform a guided random walk are much more likely to stumble across an optimum the closer they are to it, and will do so more quickly.
Therefore I believe that a dataset not only _influences_ the performance of algorithms, but is actually _an essential part of the definition of an optimization problem_.
Therefore, any benchmark must at the very least standardize or control the available data, and ideally deliberately choose the dataset to set the difficulty of the optimization problem.

Regarding standardization, I think that the authors did this reasonably well: they used the same subset of ZINC for all their baselines.
I would however like to see the authors more explicitly state that ZINC is the _only_ dataset permitted
when using the benchmark, so that future researchers cannot report spuriously high scores in the future achieved via careful dataset selection.
I think this would address the issue of standardization.

Regarding the choice of dataset, the authors chose to use ZINC because of its "pharmaceutical relevance, moderate size, and popularity" (line 201).
While ZINC does have all these properties, I think that the authors could have chosen their dataset more intentionally:
for example, they authors could imitate the Guacamol paper and choose the dataset more deliberately to influence the difficulty of the problem.
In Guacamol, the authors decided to make the tasks "exploratory", and filtered their training dataset to remove molecules with high objective function values or molecules too similar to the molecules which the objective functions were based on, so that any algorithm would need to explore molecules far away from the training set to receive a good score.
I think that the authors should make a similar explicit choice (either the same as Guacamol or perhaps testing "exploitative" algorithms instead),
and choose the dataset accordingly (if exploratory behaviour is desired then using the Guacamol dataset itself might be a wise choice).
Barring this, as a minimum I think the authors should have examined ZINC more closely with respect to the objective functions chosen,
for example performing analysis of the distribution of objective function values on ZINC for all objectives, and perhaps some post-hoc analysis of the molecules found during optimization to see how closely they match ZINC.

Overall, I think the dataset and its impact are under-discussed and under-examined.

**Choice of objective functions**

The authors based their benchmark off of a suite of previously published objective functions, chiefly Guacamol, but with some random-forest/SVM models too. Although the high number of objective functions may give the appearance that a wide variety of functions are being tested, I believe this is not the case based on my experience working with these exact functions.

- Guacamol functions are almost all based on fingerprint similarity to a real-world drug molecule, and therefore are generally 1) fairly smooth w.r.t. changes in molecular structure 2) unimodal 3) amenable to fairly strong performance from random datasets of generally drug-like molecules. These functions are very easy to optimize with random-walk like algorithms, as there tends to be a smooth path from any starting molecule towards the optimum via a series of small molecular changes which monotonically increase the objective function value. While some real-world drug design objectives resemble this (e.g. optimizing an initial "hit" molecule), others do not (e.g. finding an initial hit molecule, which is almost always a multi-modal function).
- Model-based oracles (JNK3/GSK3b/DRD2) suffer from what I would call a "fragment bias": because they are trained on extremely sparse fingerprint vectors (i.e. vectors representing the counts or presence of certain substructures), in my experience there tends to be a simple relationship between certain structures and the predicted score (i.e. some structures increase the score while others decrease them). Therefore these functions tend to give a high score to extremely large molecules which have many copies of "positive" substructures and few copies of "negative" substructures. I think this is unnaturally well-suited to "fragment-based" algorithms which can learn which fragments are helpful/harmful and then add helpful fragments.
- QED is generally very easy to optimize: table 4 shows that almost every algorithm achieves similar performance (including random screening) with minimal variation.

My suggestions to the authors would be:

1. Include some more diverse functions not based on fingerprints (perhaps something based on docking? a few docking oracles have been released in the past year)
2. Re-balance the functions to have at most 1/3 based on fingerprint similarity
3. Exclude QED
4. Try to categorize functions as "exploratory" vs "exploitative" based on the strategy needed to optimize them, and try to balance the benchmark between exploratory and exploitative functions.

**Baselines**

Although I am pleased that the authors attempted to investigate a large number of baselines, I believe that there are some issues with the way that the authors have implemented them.

The most widespread issue is the way in which each algorithm uses the starting dataset. In addition to my thoughts on the importance of the starting data, I have 2 concerns about the way the baseline algorithms use this data.

1. Extremely similar algorithms use the starting data differently. For example, consider genetic algorithms, whose behaviour depends strongly on the starting population in a similar way (see discussion of "random walk" algorithms above), and have similar failure modes (e.g. insufficient diversity in the population leading to lack of progress). Despite this, all GAs tested seem to choose their starting population differently: STONED uses 500 random smiles, Synnet uses ~200 random smiles (can't tell exactly how many), SMILES GA uses 400 random smiles, SELFIES GA uses just the single smiles "C", Graph GA uses 120 random smiles, and DST uses 2000 random smiles. Surely some of the performance differences between these GAs results from this, instead of the different mutation operations or population selection mechanisms. I think this introduces unnecessary bias towards certain algorithms, and moreover the random selection of smiles introduces an extra source of variation. I think that the starting smiles should be identical between different algorithms (perhaps chosen with the same random seed) to eliminate this source of random variation.
2. Often the chose method of using the data is sub-optimal. In my experience with GAs for molecules, I think that the best method is generally to use a large fraction of the evaluation budget to randomly evaluate molecules (at least 10%), which is used to create the initial population for the GA (e.g. selecting the best molecules, or a diverse subset). Only DST seems to do this, the rest do something which I believe is worse. This would explain why some GAs do not exceed the performance of screening on certain tasks in Table 4. This also more closely matches the original Guacamol implementations, which do a large amount of random screening before starting the GA (often they start from the top few hundred molecules after randomly choosing several tens of thousands).

A second issue is early stopping, which the authors apply to many methods. I think that this is bad practice. Stochastic algorithms often proceed via random, low-probability jumps. If the probability of making the necessary jump is sufficiently low then it will take many attempts until the jump is made. In my experience using a variety of these algorithms, there are often long periods of no progress before a sudden increase. Early stopping removes the possibility of seeing this. One particular example where I have seen this in my own experiments is GraphGA (try running it for ~10x as long as the original Guacamol paper and you will see that the performance does not really stop increasing and is much higher than the performance in the table of that paper). By doing early stopping, I think that the authors' results systematically understate the true performance.

A third issue is batch sizes: many of the algorithms use very large batch sizes, which gives fewer opportunities to iteratively improve on the molecules found. For example, DoG-Gen has a batch size of several thousand, meaning it only has ~3 opportunities to iterate on a molecule, while other algorithms have perhaps 10x as many iterations. I would like to see the authors:
1. Choose the batch size consistently, at least within algorithms of the same class. Not doing this just introduces an additional source of bias.
2. Use small batch sizes where possible, since this will generally lead to better performance given a fixed number of calls to $f$

A fourth issue: I think there may be a bug in your code for screening. I downloaded your version of the ZINC dataset and computed the objective function for a random subset of 100k molecules for Zaleplon MPO (where screening does poorly). It looks like the 99th percentile in the dataset is ~0.39, which makes the AUC of 0.072 seem surprisingly low. You should double check that this is implemented correctly. In my experience, screening can be a frustratingly strong baseline.

Overall, I have serious concerns that at least half of the baselines are are set up in a highly sub-optimal way.
I understand that implementing all baselines optimally might require the authors to be an expert in all the baselines, which is a lot of work; the authors acknowledge this in their conclusion (line 306).
It may also seem like I am just asking the authors to "do more tuning" which is a common request from conference reviewers almost regardless of how much tuning was done.
However, I feel in this case that asking for better tuning of the baselines is justified because:
1. Some key parameters were not tuned at all, and as an expert familiar with these methods I believe that their default setting is poor.
2. I think it is important to have very strong baselines when proposing a new benchmark to ensure that it is actually challenging to achieve SOTA performance.
3. Disregarding how good the tuning was, things like batch sizes and starting datasets are _inconsistent_ between different algorithms, which makes comparing them less meaningful. Comparing algorithms is another important purpose of baselines.
4. The baselines are, in my opinion, the biggest contribution of this paper, and therefore should be held to the generally high standard that the NeurIPS D&B track has set for itself.

**Documentation:**

Good; it is hosted on github and all code is available. In general I found reviewing the code to be quite useful!

**Ethics:**

No concerns.

**Relation To Prior Work:**

I think that this paper cites and discusses all prior work appropriately. One suggestion I have is to analyze the commonalities between different algorithms more, rather than treating all algorithms as a fixed set of design choices. For example, all GAs have populations, and mainly differ in how mutations occur. REINVENT and hill-climbing are essentially just 2 different ways of training the same policy. I think if you do this the class of algorithms that you consider might be more general and clearer to the reader.

**Summary And Contributions:**

**My summary of the paper**: This paper proposes a benchmark for molecular optimization (i.e. optimizing a function $f$ over the space of molecules). Noting that real world functions $f$ are either real-world experiments or expensive simulations, the proposed PMO benchmark tests _sample-efficiency_ of molecular optimization algorithms (i.e. can they produce optimized molecules with few calls to $f$). Specifically, the benchmark consists of a new set of metrics for a set of previously published objective functions (from Guacamol and TDC) to optimized with at most 10k function calls. The authors evaluate a relatively "out of the box" version of 25 different algorithms (with a bit of tuning) and show that 1) performance is much lower in the limited sample regime and 2) recent "state-of-the-art" algorithms don't actually seem to do any better than established older algorithms in this regime.

The authors don't make an explicit list of their contributions in the paper, but I would state their contributions as:
1. Selection of 23 objective functions + metrics to define the PMO benchmark
2. Evaluation and comparison of 25 previously-published algorithms on the PMO benchmark.
3. Several consequential conclusions from these experiments, chiefly that GraphGA and REINVENT seem to generally outperform most other (more complex and more recently published) methods

**Summary of my review**: In general I liked the work and think it is a step in the right direction: sample-efficiency is very important and very overlooked.
Reading the introduction to the paper, my understanding is that the authors propose PMO partly to stop the flow of papers which test their algorithms on unrealistic tasks and compare to very weak baselines, concluding from this that their method is "state of the art" despite being nearly worthless in practice.
My overarching perspective when judging this work is "does PMO achieve this goal".
Taking this and the scope of NeurIPS D&B into account, and given that this work
does not propose new algorithms, objective functions, or datasets,
I think that the relevant questions for evaluating this work are:
1. Are the proposed metrics and setup of PMO good choices?
2. Is the specific choice of objective functions in PMO good?
3. Are the experiments with baseline algorithms meaningful and correct? Will the reported results hold new algorithms using the PMO benchmark to a high standard?

My opinion on these criteria are 1) metrics/setup make sense, although the choice of 10k is somewhat arbitrary 2) objectives are ok but inherit biases from Guacamol 3) some baselines are good, but I worry that many of them are implemented sub-optimally which understates their potential performance.
Ultimately I think that PMO is a promising first step towards a better molecular optimization benchmark but only incrementally improves upon existing benchmarks.
My opinion therefore is to recommend against acceptance at this time, although I believe that a revised version of PMO could potentially be accepted to a venue like D&B in the future.

---

> ### Author Response · Authors · 2022-08-15
> **Response to comments from reviewer i48g**
>
> Thank you for your comments. Please find our response below. We have uploaded more results in the Appendix and highlighted the important changes in the draft in red.
>
> **Q1**: Metrics
>
> **A**: Thanks for agreeing with our using AUC as metrics. Regarding top-10, it aligns closer to the actual drug discovery scenario than top-1, as we will always keep a handful of candidates in the preclinical stages to reduce the risk of failure in clinical trials. We have moved Table 3 (with multiple metrics like top-1 and AUC top1 in it) to the main text, and an evaluation of synthetic accessibility and diversity can be found in Appendix A.3 and A.4.
>
> **Q2**: 10k budget
>
> **A**: There are multiple types of experiments, but most real wet-lab experiments and high-fidelity simulations could be on the scale of 10^2 to 10^3 (with a reasonable academic lab budget). As we are using AUC, we just need to have this 10k as an upper bound: AUC can continue to reward good early performance so that it can distinguish between methods that do well with just hundreds or even tens of (hypothetically) evaluations.
>
> **Q3**: Choices of dataset
>
> **A**: Thanks for agreeing with our choice of dataset. We have now re-emphasized it in section 3.1 to avoid misunderstanding.
>
> **Q4**: Choice of objective functions
>
> **A**: Please see our response to Reviewer VZwE’s question regarding current oracles and docking. We keep QED in our test to demonstrate that it is trivial. The “exploratory” and “exploitative” classification doesn’t have a clear boundary as some extent of “exploration” is always needed.
>
> **Q5**: Baseline
>
> **A**:
> - We have tuned a reasonably large hyperparameter space for most methods and visualized more tuning results in Appendix D.2. Note this is still not a complete result as there are too many tuning results to show all of them. The batch sizes and the number of the initial evaluation for genetic algorithms are results of hyper-parameter tuning. We believe that experimental results are more sound than intuition. In addition, PMO is open-sourced and accepts community contributions. As long as one can show another set of hyperparameters is better overall, we would like to replace it.
> - Regarding early stopping, please see our response to Reviewer RkUE’s **M3**. Every algorithm can reach a higher value if one runs it longer. But our focus is on determining which algorithm could do better given a limited evaluation budget.
> - We have screened the ZINC 250k and visualized the distribution of Zaleplon MPO values in Figure 31. As you can see, the highest value in the dataset is about 0.39, and most of the molecules are around 0. If you can find a bug in our code, we would like to fix it.

---

> > ### Comment · Reviewer_i48g · 2022-08-18
> > **Bug in code for Zaleplon**
> >
> > > We have screened the ZINC 250k and visualized the distribution of Zaleplon MPO values in Figure 31. As you can see, the highest value in the dataset is about 0.39, and most of the molecules are around 0. If you can find a bug in our code, we would like to fix it.
> >
> > I think that the TDC's implementation of Zaleplon does not exactly match that of the original Guacamol package. I compared values on a set of molecules and found several mismatches. My guess is there is a mistake copying over the code from Guacamol. I would raise a Github issue, but that would break anonymity. Please look into the issue, perhaps make a PR on TDC's github, and link it here in the comments for me. I would be happy to consider the issue resolved once I see this.

---

> > > ### Author Response · Authors · 2022-08-27
> > > **No bugs in our code**
> > >
> > > Thanks for your comments. We reply to a few points here:
> > >  - We conducted the hyper-parameter tuning for methods in our benchmark with our AUC Top-10 metrics. Please see D.2 for a visualization of the results of tuning and B.1 for setups.
> > >  - The implementation of TDC's Zaleplon doesn't count Hydrogens, which still makes it a valid oracle and doesn't change the quality of our benchmark. We added, "Note that the implementation of sitagliptin\_mpo and zaleplon\_mpo are different from the ones in Guacamol" and made changes to add original versions of the oracles (https://github.com/mims-harvard/TDC/commit/d735525af72c0bf2574f6e1d1afd8d9fba72ffef).

---

> > > > ### Comment · Reviewer_i48g · 2022-08-28
> > > > **Thanks for addressing bug**
> > > >
> > > > Thanks for changing/adding this. I think it would be confusing to have 2 oracles with the same name that are actually different functions. Glad that you have clarified/changed this.

---

> > ### Comment · Reviewer_i48g · 2022-08-18
> > **Responses to some points**
> >
> > Thank you for your response. I will reply to a few select points here:
> >
> > > Regarding top-10, it aligns closer to the actual drug discovery scenario than top-1, as we will always keep a handful of candidates in the preclinical stages to reduce the risk of failure in clinical trials.
> >
> > Yes, but if all 10 candidates are similar this does not significantly reduce the risk of failure in clinical trials, since the outcomes are likely to be correlated.
> >
> > > Regarding early stopping, please see our response to Reviewer RkUE’s M3. Every algorithm can reach a higher value if one runs it longer. But our focus is on determining which algorithm could do better given a limited evaluation budget.
> >
> > I agree on the focus, my concern is that _you do not give these algorithms the full evaluation budget_. Your response to the other reviewer did not seem to address this directly.
> >
> > > The batch sizes and the number of the initial evaluation for genetic algorithms are results of hyper-parameter tuning.
> >
> > Can you be more specific: did you do this hyperparameter tuning, or did the authors of the original papers? One potential problem is that if the authors tuned their methods on tasks like logP/QED, their tuned parameters may not work well for other objective functions.
> >
> > > We believe that experimental results are more sound than intuition.
> >
> > I think this is a mis-characterization of my position 😛 . Obviously intuition should be informed by experiments, but I think it is also not right to generalize from limited observations (e.g. experiments in one setting within a narrow range of hyperparameters). With benchmarking, ultimately I think nobody cares what the results are on these specific tasks (e.g. Guacamol tasks are all artificial), what we care about is choosing algorithms to perform well on _new_ tasks. Judging this in my opinion requires intuition, since we don't have anything better at the moment.

---

### Official Review · Reviewer_5BVJ · 2022-07-27
**A benchmark with significance but a little too simple**

**Rating:** 6
**Confidence:** 4
**Clarity:** The paper is written well.

**Strengths:**

1. This paper integrates existing molecular optimization algorithms using a unified framework and then aligns them for comparison. The overall workload is large, and the work has practical significance.
2. This paper considers the oracle budget problem in more practical scenarios of molecular design, which has been ignored in previous work.
3. The writing of the article is good, and the code is clear.


**Weaknesses:**

1. This article only considers the evaluation on single objective in each time. However, practical drug development is usually a multi-objective optimization, such as considering both efficacy and synthesis route. So, the “practical” in the paper is a bit over claimed.
2. Why is the number of oracle calls important? Since oracle calls can also be calculated, why limit it to 10000?


**Additional Feedback:**

In practical sense, multiple objectives should be considered in molecule optimization.

**Correctness:**

The evaluation method of the paper is correct, but can be improved to be more in line with the actual development of drugs. For example, more objectives are considered at the same time in a single optimization.

**Documentation:**

Yes, the authors have open-sourced the code.

**Ethics:**

No.

**Relation To Prior Work:**

This article clearly illustrates the relationship with the previous work. That is, the efficiency problem in actual drug development is considered and oracle calls is limited. The molecular optimization algorithms are compared under the same criteria.

**Summary And Contributions:**

This article presents a benchmark for molecule optimization to facilitate the transparent and reproducible evaluation of algorithmic advances in molecular optimization, consisting of 25 molecular optimization algorithms and 23 tasks with a particular focus on sample efficiency.  Compared with previous work, this benchmark takes into account the limits in practical scenarios of molecular design. The final experimental results also yielded some interesting conclusions.

---

> ### Author Response · Authors · 2022-08-15
> **Response to comments from reviewer 5BVJ**
>
> Thank you for your comments. Please find our response below. We have uploaded more results in the Appendix and highlighted the important changes in the draft in red.
>
> **Q1**: This article only considers the evaluation of a single objective each time. However, practical drug development is usually a multi-objective optimization, such as considering both efficacy and synthesis route. So, the “practical” in the paper is a bit over claimed.
>
> **A**: We added a sentence to acknowledge the current results are about single scalar objectives in the abstract. It is worth noting that  a majority of reports of “multi-objective optimization” in molecular generation are simply recast into single-objective ones by scalarizing the objective (e.g., taking the sum or product). In this setting,  the requirements for optimization algorithms are identical. Our benchmark also sets a basis for multi-objective optimization, and we plan to extend the codebase to include  multi-objective optimization in due time.
>
> **Q2**: Why is the number of oracle calls important? Since oracle calls can also be calculated, why limit it to 10000?
>
> **A**: As mentioned in the introduction, one can calculate the toy oracle functions much more times, but most valuable oracles---experiments or high-accuracy simulations---require substantial costs. If we want to apply a molecular optimization algorithm to optimize such a valuable oracle in realistic discovery scenarios, it is vital to identify the desired compound with as few oracle calls as possible.

---

### Official Review · Reviewer_wkmP · 2022-07-27
**PMO review**

**Rating:** 6
**Confidence:** 3

**Strengths:**

(1)	Reasonable evaluation metric taking sample efficiency into consideration with clear motivation;

(2)	Sample efficiency is very important for molecular lead optimization in real applications, this reviewer values the significance of this contribution very much;

(3)	The organization of current methods is systematic and logical;

(4)	Re-benchmarking leads to some important empirical findings and new insights;

(5)	Significant contributions for further developments of molecular optimization and lead to real impacts on pharmaceutical industries in the future.


**Weaknesses:**

This paper does not propose new dataset. As admitted by the authors, they did not exhaustively explore every method and thoroughly tune every hyperparameter.  Some other important quantities are not measured.

**Additional Feedback:**

No

**Clarity:**

Yes, this paper is well written. Clear motivations for new benchmark proposal, reasonable evaluation metrics design.

**Correctness:**

Yes. The claims made in the submission is correct. The evaluation method proposed is very reasonable. This reviewer agrees that sample efficiency is truly an important and overlooked evaluation metric.

**Documentation:**

Yes

**Relation To Prior Work:**

Yes, this work clearly points out the drawbacks of previous works and proposes corresponding solutions.

**Summary And Contributions:**

This work proposes a new evaluation metric balancing with sample efficiency, inspired by real application scenarios. With new evaluation metric, this work re-benchmarks all proposed methods and discovers that the traditional methods are still very competitive and SMILES-based representation of molecules do not have convincing deficiencies.

This paper has clear motivation and proposes a significant overlooked issue in molecular optimization. Then the proposed corresponding evaluation metric is reasonable and intuitive. The survey and taxonomy construction for molecular optimization methods are comprehensive and systematic. Re-benchmarking all proposed methods is a great effort for this community.

---

> ### Author Response · Authors · 2022-08-15
> **Response to comments from reviewer wkmP**
>
> Thank you for your comments. Please find our response below. We have uploaded more results in the Appendix and highlighted the important changes in the draft in red.
>
> **Q1**: This paper does not propose new dataset.
>
> **A**: That is correct; this is a benchmarking paper but not a dataset paper, but not proposing new datasets should not be a weakness in this track.
>
> **Q2**: As admitted by the authors, they did not exhaustively explore every method and thoroughly tune every hyperparameter.
>
> **A**: We believe no paper can truly exhaustively explore every method and thoroughly tune every hyperparameter. We followed a rigorous hyperparameter optimization protocol that Reviewer RkUE called “near-exhaustive”.
>
> **Q3**: Some other important quantities are not measured.
>
> **A**: An evaluation of synthetic accessibility and diversity can be found in Appendix A.3 and A.4.

---

### Official Review · Reviewer_RkUE · 2022-07-28
**Excellent Benchmark, Concerns with Claims**

**Rating:** 7
**Confidence:** 5

**Strengths:**

&nbsp;

1. The empirical evaluation is near-exhaustive.

2. The benchmark is informative. Of particular interest is the finding that older models such as REINVENT are performant when benchmarked on new datasets. I believe this is an important message for the molecule generation community.

3. The benchmark focusses on key indicators of performance in real-world molecular discovery campaigns such as sample efficiency.

&nbsp;

**Weaknesses:**

&nbsp;

## **MAJOR POINTS** ##

&nbsp;

1. The early stopping criterion of 5 iterations mentioned in appendix B3 seems to be far too harsh, especially for methods such as Bayesian optimisation which can often plateau during the optimisation trace due to exploratory actions. Most importantly, the results from [1], who also use the Therapeutics Data Commons benchmark show that various VAE BO methods [1,2,8,9] achieve the strongest performance and also obtain different orderings in terms of the baseline models. It would be great to see the performance comparison in terms of the raw scores (as opposed to the AUC) between [1] and the current submission.

&nbsp;

2. In terms of the statement,

&nbsp;

> Most papers do not report how many times the oracle function is called to achieve the reported results (i.e. how many candidate molecules were evaluated), except in rare cases

&nbsp;

It would be great if the following papers were also cited as the rare cases! [1, 2]

&nbsp;

3. In terms of the claim,

&nbsp;

> Our results show that none of the existing molecular optimization algorithms are efficient enough to solve a de novo molecular optimization problem within a realistic oracle budget of hundreds of experiments, and “state-of-the-art” methods often fail to outperform their predecessors

&nbsp;

I am not convinced that the authors yet have sufficient empirical evidence to make this claim given the conditions under which the models were run e.g. the early stopping of 5 iterations without performance improvement for Bayesian optimisation methods. I would recommend moderating the claims as they could potentially discourage research into otherwise promising methods. I don't believe moderating the claims takes away from the contributions of the paper, the majority of benchmarks are submitted without extensive empirical evaluation and the authors have gone above and beyond the remit in this regard.

&nbsp;

4. In terms of the statement,

&nbsp;

> but this is not an issue for more recent methods. We believe this is because current language models are better able to learn the grammar of SMILES strings, which has flattened the advantage of SELFIES.

&nbsp;

One of the reasons why the original SMILES VAE struggled to generate valid outputs was shown empirically in [3], namely a disconnect between the VAE and the Bayesian optimisation scheme. More specifically, the exploratory component of the acquisition function encourages points to be queried which lie far from the training data manifold in latent space. The constrained Bayesian optimisation approach introduced in [3] alleviated some of the problems with SMILES validity by learning a feasible (high validity) region of VAE latent space. As such, the validity concerns with the SMILES grammar are unlikely to be solely due to the quality of the language model, at least in the case of the VAE models, and I believe this is worth mentioning!

&nbsp;


## **MINOR POINTS** ##

&nbsp;

1. It would be great if the references appeared in numbered order e.g. via the LaTeX command:

&nbsp;

> \bibliographystyle{unsrtnat}

> \setcitestyle{numbers,open={[},close={]},citesep={,}}

&nbsp;

2. For the GP BO method, in addition to Morgan fingerprints, "fragprints" [4] i.e. concatentations of RDKit fragment vectors (count vectors over functional groups in the molecule) with the Morgan fingerprint bit vectors have been found to outperform Morgan fingerprints in the prediction of transition wavelengths. Given that the choice of featurisation is open-ended, frapgrints way be worth mentioning as an alternative choice for some tasks. The combination of the "local" fingerprint features and the "global" fragment features appears to perform well over a range of MoleculeNet tasks.

&nbsp;

3. The batch size for BO mentioned in the appendix (1,180) is very large relative to the optimisation horizon (10,000). Presumably the batch size is too high due to the scalability issues? Is this for an exact GP?

&nbsp;

4. In terms of the statement on line 41,

&nbsp;

> Further, some papers only report results on trivial oracles [33] like quantitative estimate of drug-likeness (QED) or penalized octanol-water partition coefficient (LogP)

&nbsp;

It might be an idea to expand on the meaning of trivial here. Maximising the penalized octanol-water coefficient is not meaningful from the point of view of drug discovery yet it may act as a toy task for differentiating machine learning architectures based on their performance. In contrast, while QED has some relation to drug-likeness, the metric is generally too easy to be used for model comparison.

&nbsp;

5. In Table 1, [1] could be added in the entry corresponding to VAE + BO + SELFIES. This work uses a transformer architecture which appears to be more powerful than the RNN/GRU implementations of the SMILES VAE from Gomez Bombarelli et al. and Polykoskiy et al.

&nbsp;

6. In section 2.1 it is probably worth explicitly referencing the auxiliary properties of black-box optimisation problems [5] e.g. while the analytic form of the function is unknown, the function may be queried pointwise, the gradients of the function are inaccessible and there is a high cost (financially in addition to time) incurred by querying the black-box. In addition, it might be worth describing the nature of oracles in laboratory synthesis, where querying a (high fidelity) experimental property is much more time-intensive relative to computing a (lower fidelity) property value *in silico*.

&nbsp;

7. I would recommend using a different Greek letter for the design space since it bears a resemblance to the symbol for the oracle.

&nbsp;

8. In terms of the statement

&nbsp;

> Two-dimensional (2D) graphs can intuitively define molecular identities to a first approximation (ignoring stereochemistry)

&nbsp;

Although this is a widely-cited limitation of 2D molecular graph representations, it may be worth omitting that 2D graphs ignore stereochemistry since in theory, it should be possible to incorporate stereochemical information through node/edge attributes and graph transformations [6].

&nbsp;

9. In terms of the statement

&nbsp;

> Screening (a.k.a. virtual screening) involves searching over a pre-enumerated library of molecules exhaustively. We include Screening as a baseline, which randomly samples ZINC 250k.

&nbsp;

It might be worth removing "exhaustively" since this seems to be at odds with "random sampling" in the next sentence.

&nbsp;

10. In terms of the use of MolPAL, it may be worth mentioning that many other choices of surrogates could be used.

&nbsp;

11. In relation to the statement,

&nbsp;

> We included a string-based model, BO over String Space (BOSS) [32], and a synthesis-based model, ChemBO [27], but do not obtain meaningful results even with early stopping

&nbsp;

It is probably worth citing the poor scaling of the SSK kernel in particular here as the reason and pointing to sparse implementations as a potential solution.

&nbsp;

12. Reference 26, Auto-encoding variational Bayes was published at ICLR 2014.

&nbsp;

13. It would be great if BoTorch [7] was cited given that is used for the experiments.

&nbsp;

14. In section B.4 of the appendix,

&nbsp;

> Worth to mention that due to the nonparametric essence, Bayesian optimization methods scales poorly with the data size, the run process is notoriously and intolerably slow even dealing 1k training data

&nbsp;

This is likely because the authors are using exact GP implementations in the codebase as far as I can see for both GP BO and the SMILES VAE. [1-3, 8] as well as the original automatic chemical design paper by Gomez-Bombarelli et al. all use sparse GPs for scalability. On the other hand, BOSS, at the current point in time, is constrained to be an exact GP as there is no sparse implementation of the SSK string kernel. The exact version of this algorithm indeed scales very poorly.

&nbsp;

15. It would be great if Adam was cited given that it is used [10].

&nbsp;

16. Line 594, typo, "determinantal point process".

&nbsp;

17. Some typos in the references e.g. missing capitalisation for "Bayesian" and missing capitalisations after colons in the titles of papers.

&nbsp;

## **REFERENCES** ##

&nbsp;

[1] Maus et al. Local Latent Space Bayesian Optimization over Structured Inputs. arXiv, 2022.

[2] Grosnit et al. High-Dimensional Bayesian optimisation with Variational Autoencoders and Deep Metric Learning. arXiv, 2021.

[3] Griffiths and Hernández-Lobato, Constrained Bayesian Optimization for Automatic Chemical Design using Variational Autoencoders. Chemical Science, 2020.

[4] Thawani et al. The Photoswitch Dataset: A Molecular Machine Learning Benchmark for the Advancement of Synthetic Chemistry. arXiv 2020.

[5] Garnett, Bayesian Optimization, Cambridge University Press, 2022.

[6] Andersen et al. Chemical Graph Transformation with Stereo-Information. In International Conference on Graph Transformation, 2017.

[7] Balandat et al. BoTorch: A Framework for Efficient Monte-Carlo Bayesian Optimization. NeurIPS, 2022.

[8] Tripp et al. 2020. Sample-Efficient Optimization in the Latent Space of Deep Generative Models via Weighted Retraining. NeurIPS, 2020.

[9] Eriksson et al. Scalable Global Optimization via Local Bayesian optimization. NeurIPS, 2019.

[10] Kingma and Ba, Adam: A Method for Stochastic Optimisation, ICLR, 2015.

&nbsp;

**Additional Feedback:**

&nbsp;

Adequately addressed.

&nbsp;

**Clarity:**

&nbsp;
The paper is excellently written and presented.
&nbsp;

**Correctness:**

&nbsp;
cf. main response
&nbsp;

**Documentation:**

&nbsp;

The documentation of the codebase is excellent.

&nbsp;

**Ethics:**

&nbsp;

Adequately addressed.

&nbsp;

**Relation To Prior Work:**

&nbsp;

cf. main response

&nbsp;

**Summary And Contributions:**

&nbsp;

The authors provide an extensive benchmark for molecular property optimisation together with a near-exhaustive empirical evaluation of molecule generation architectures. I believe this project is timely, well-needed and that the NeurIPS Datasets and Benchmarks track is an excellent venue for this work.

I do however have concerns about some of the claims made in the submission, particularly with respect to the performance of the BO and VAE-BO methods in light of recent work [1, 2]. As such, my evaluation at this stage is borderline as it is unclear whether it will be possible to address all of the issues in the time allocated for author response.

If however the authors can address the issues adequately, I will be willing to substantially raise my score and argue for the acceptance of this paper. I believe this will be a very important and impactful benchmark and it would be a great shame if the paper is not published in either this round or the next round of NeurIPS Datasets and Benchmarks.

In terms of facilitating communication, it would be great if the authors could reply (as much as is possible) with partial responses to the points raised as opposed to providing the full author response at the end of the discussion period. The latter approach leaves little time to review the response and update the score.

&nbsp;

---

> ### Author Response · Authors · 2022-08-15
> **Response to comments from reviewer RkUE (1/2)**
>
> Thank you for your comments. Please find our response below. We have uploaded more results in Appendix and highlighted the changes in the draft in red. Capital M refers to “major” and lower case “m” refers to “minor.” "A" refers to "Answer."
>
> **M1**: BOs & early stopping.
>
> **A**: The results of molecular optimization can be found in https://figshare.com/articles/dataset/Results_for_practival_molecular_optimization_PMO_benchmark/20123453, and one can see that the early stop strategy doesn’t affect SMILES/SELFIES VAE BO in most cases (they reach 10k oracle calls). JT VAE BO has early stopped more often but each run reached at least 1,291 oracle calls, which is a reasonable number to draw meaningful conclusions. We have incorporated one method you mentioned to achieve the strongest performance into our benchmark (https://github.com/huawei-noah/HEBO/tree/master/T-LBO, with JTVAE as the VAE model following the original paper) and evaluate its results with the default configurations on two oracles in Figure 1 (isomer_c9h10n2o2pf2cl and celecoxib_rediscovery). Though the computational overhead is prohibitively expensive (~17hrs for 500 oracle calls) that we cannot tune the hyperparameter and run a thorough evaluation, we observe that T-LBO’s performance is close to other VAE-BO methods. The results (AUC Top-10) are attached below:
>
> |                | celecoxib\_rediscovery | isomers\_c9h10n2o2pf2cl |
> | --------- | ---------- | ----------- |
> | GP BO | 0.723 | 0.469 |
> | DoG-AE | 0.355 | 0.049 |
> | SMILES VAE | 0.354 | 0.084 |
> | SELFIES VAE | 0.326 | 0.200 |
> | JT VAE | 0.299 | 0.090 |
> | **JT VAE + T-LBO** | **0.257** | **0.013** |
>
> **M2**: Additional rare cases reported number of oracle calls.
>
> **A**: Thanks for the advice. We have added these references and others.
>
> **M3**: Moderating the claims, primarily due to the use of early stopping for BO methods.
>
> **A**: Please see our response to **M1** that the current results of VAE BOs are all above a thousand evaluations, and that’s enough to validate our conclusion that the methods cannot optimize within several hundreds of evaluations. We also note that the early stopping is added for various methods, not only BOs. Because all of the methods are stochastic to some extent, we should always expect to see an improvement if we wait long enough time. Overall, PMO is a live leaderboard that accepts community contributions. We have provided a detailed guide on how to add a new method into our codebase and benchmark it (https://github.com/wenhao-gao/mol_opt/blob/main/CONTRIBUTE.md). We believe the current results would encourage researchers to develop more efficient methods rather than discourage them.
>
> **M4**: Another reason original SMILES VAE struggled to generate valid outputs
>
> **A**: We observed the improvement in validity not only in SMILES VAE and but also in other methods like REINVENT and SMILES LSTM HC that don’t use a BO. Besides, we are using a naive BO but not the constrained BO proposed in the paper you mentioned. Therefore, we still see this as evidence that  the more powerful language models are the main reason for this improvement.
>
> **m1**: The references in numbered order
>
> **A**: Thanks for the advice. We have changed the style.
>
> **m2**: For the GP BO method, in addition to Morgan fingerprints, "fragprints" i.e. concatentations of RDKit fragment vectors have been found to outperform Morgan fingerprints.
>
> **A**: In this paper, we tried to include a representative set covering as many types of algorithms as possible. Within each type, there are many variations of basic methods, like the constrained BO and all other latent space optimization on SMILES VAE as extensions, which contributes to the fact that we cannot exhaustively screen every method and set of hyperparameters. In terms of molecular fingerprints, there are nine types of fingerprints available in RDKit [1], and more elsewhere such as fragprints [2], or MAP4 [3].
>
> **m3**: The batch size for BO mentioned in the appendix (1,180) is very large relative to the optimisation horizon (10,000). Presumably the batch size is too high due to the scalability issues? Is this for an exact GP?
>
> **A**: This value is a result from a hyper-parameter tuning, which one can find more details in the section D.2 in Appendix. We acknowledge it might not be a `best` value but it is part of the best set of parameters among a reasonably large parameter space we explored. Yes, this is an exact GP and we directly adopted it from its original implementation.
>
> **m4**: Definition of trivial in oracle.
>
> **A**: We have added footnotes in the draft to explain why those oracles are trivial.
>
> **m5**: Additional VAE + BO + SELFIES example.
>
> **A**: There are always more examples to add, so in this launch of the benchmark  we aimed to list a representative set of methods, instead of an exhaustive one. We did look at the paper you mentioned, but it is neither published (reviewed), nor open-sourced (at the time we saw you comment).

---

> > ### Comment · Reviewer_RkUE · 2022-08-24
> > **Reviewer Response**
> >
> > &nbsp;
> >
> > Many thanks to the authors for their detailed response and I apologise for my own delay in responding to their comments!
> >
> > &nbsp;
> >
> > **M1**: I appreciate the time and effort the authors have given towards adding an additional model to the benchmark. I am still not convinced about the early stopping criterion however. I have worked with Bayesian optimisation methodology extensively and early plateaus in the optimisation traces are rarely cause to terminate the algorithm. Without seeing the results for the full evaluation budget it is impossible to draw conclusions for this particular benchmark. I await the authors' response to Reviewer i48g for further discussion on this point.
> >
> > &nbsp;
> >
> > Nonetheless, I wish to highlight the authors' later point that the benchmark is designed for community contribution. It is a highly challenging task to perform exhaustive evaluation on such a diverse set of algorithms, many of which, such as Bayesian optimisation and genetic algorithms (with which Reviewer i48g seems to have extensive experience), require careful design. I think that one solution for the current paper, rather than redesign and rerun algorithms ad nauseum is to explicitly acknowledge the limitations involved in the evaluation.
> >
> > &nbsp;
> >
> > The main problem I see at the moment is that some of the scientific claims are not fully supported by the empirical evaluation (e.g. with limitations such as early stopping).
> >
> > &nbsp;
> >
> > One solution would be to moderate the claims that are not fully supported and instead leave further investigation for the community. I do not think that this would detract from the paper's impact and I believe with some rewriting the paper could be a strong contribution to the track.
> >
> > &nbsp;
> >
> > **M3**: I await the authors' response to the same point raised by Reviewer i48g.
> >
> > &nbsp;
> >
> > **M4**:
> >
> > >Therefore, we still see this as evidence that the more powerful language models are the main reason for this improvement.
> >
> > &nbsp;
> >
> > The sentence in the paper states:
> >
> > &nbsp;
> >
> > >We do observe some early methods like the initial version of SMILES VAE [6] (2016) and ORGAN [31] (2017) struggle to propose valid SMILES strings, but this is not an issue for more recent methods. We believe this is because current language models are better able to learn the grammar of SMILES strings, which has flattened the advantage of SELFIES.
> >
> > &nbsp;
> >
> > This sentence needs to be altered as it has been empirically demonstrated that language models are not the ONLY reason for the poor validity of the initial SMILES VAE.
> >
> > &nbsp;
> >
> > **m2**: I agree with the authors, an exhaustive evaluation of all such representations is infeasible and it was not my intention to suggest it. However, it may be worth mentioning the choice of representation e.g. choices of fingerprints as a determinant of performance.
> >
> > &nbsp;
> >
> > **m3**: The batch size isn't typically a tuning parameter in BO. This point would appear relate to the previous point that some algorithms require substantial domain knowledge to run. Given the extensive nature of the current study, I don't believe the authors should be held accountable for not being domain experts in every method they employ. It is a very challenging task and I applaud the authors for taking it on!
> >
> > &nbsp;
> >
> > **m5**: The paper referenced appeared on arXiv in January 2022. Concurrent submissions are generally taken to be 2 months before the date of submission according to the NeurIPS guidelines:
> >
> > &nbsp;
> >
> > https://neurips.cc/Conferences/2022/PaperInformation/NeurIPS-FAQ
> >
> > &nbsp;
> >
> > While it was not my intention to suggest that the authors reimplement the method, the paper should at least be mentioned. The conclusions on the efficacy of VAE-BO would appear to contradict the findings of the authors. Furthermore, the optimisation traces in this paper show clear evidence that early stopping would damage the performance of VAE BO with the SELFIES transformer. It would be great if the authors could discuss their findings in the context of these results. I think a moderation of the claims made in the paper would be sufficient in this regard. As an aside, the code for the method is now available open-source (albeit only 13 days ago):
> >
> > &nbsp;
> >
> > https://github.com/nataliemaus/lolbo
> >
> > &nbsp;
> >
> > I reiterate that I agree with the authors that it would be unreasonable to expect them to run this code during the rebuttal phase, a discussion and moderation of claims would be sufficient.
> >
> > &nbsp;
> >
> > **m12**: https://iclr.cc/archive/2014/conference-proceedings/
> >
> > &nbsp;

---

> > > ### Comment · Reviewer_RkUE · 2022-08-24
> > > **Summary of Reviewer Response**
> > >
> > > &nbsp;
> > >
> > > In summary, at this stage I believe the discussion should consider the validity of the claims made as well as the framing of the paper. I believe the benchmark is a valuable contribution and the authors should not necessarily be expected to tune each model perfectly. The problem arises when claims are made under the assumption that the models have been tuned perfectly. Given the potential value and impact I see in this benchmark I would be very happy to raise my score if the authors can make text-based changes to the paper such that each claim made is either removed or fully substantiated by empirical results.
> > >
> > > &nbsp;

---

> > > > ### Author Response · Authors · 2022-08-27
> > > > **Response to Reviewer's Response**
> > > >
> > > > Thanks for your comments and suggestions. Regarding your comments, we have changed the text and highlighted them in red. The main changes include the following:
> > > >  - We cite LOL-BO as the example of SELFIES-VAE in Table 1.
> > > >  - In the section on BO and VAE, we remind that there are other variants.
> > > >  - In the result and analysis of sample efficiency, we changed it to "none of the methods we implemented can optimize the simple toy objectives within hundreds of oracle calls under our experimental settings."
> > > >  - In the statement about SMILES validity, we changed it to "We believe this is partially because ..."
> > > >  - In the conclusions, we added "the representative methods we implement might not be the best-in-class among all possible variants."
> > > >  - In the description of GP BO (Appendix), we emphasized "It should be noted that there are other types of fingerprints, such as fragprints and MAP4, and the choice of that could be a major performance determinant."
> > > >  - We changed the citation of "Auto-Encoding Variational Bayes" to ICLR.
> > > >
> > > > If you find the moderated conclusions acceptable, we would appreciate it if you revised your score. Thank you for your time in reviewing our paper!

---

> > > > > ### Comment · Reviewer_RkUE · 2022-08-29
> > > > > **Raised Score**
> > > > >
> > > > > &nbsp;
> > > > >
> > > > > I have now raised my score and recommend acceptance. Even if the ambitious attempt at an exhaustive empirical evaluation is not flawless, I think the current work is in the spirit of the benchmarks track in so far as it acts as a starting point to direct the community towards performing the empirical evaluation themselves. I also think it is important to have such socially impactful research domains represented at the NeurIPS Datasets and Benchmarks track.
> > > > >
> > > > > &nbsp;
> > > > >
> > > > > In terms of the claim,
> > > > >
> > > > > &nbsp;
> > > > >
> > > > > > none of the existing molecular optimization algorithms are efficient enough to solve a de novo molecular optimization problem within a realistic oracle budget of hundreds of experiment
> > > > >
> > > > > &nbsp;
> > > > >
> > > > > it may be worth further clarifying this point as there are many moving parts to real-world molecular optimisation problems such as the dataset and/or auxiliary data used to train algorithms. The claim that,
> > > > >
> > > > > &nbsp;
> > > > >
> > > > > > state-of-the-art methods often fail to outperform their predecessors
> > > > >
> > > > > &nbsp;
> > > > >
> > > > > could also be moderated by using a phrase such as "pending community contributions".

---

> > > > > > ### Author Response · Authors · 2022-08-29
> > > > > > **Thank you for your suggestion**
> > > > > >
> > > > > > Thank you for the suggestion and the revision of the score. We will modify the text to highlight our benchmark's live nature further and seek community contribution.

---

> ### Author Response · Authors · 2022-08-15
> **Response to comments from reviewer RkUE (2/2)**
>
> **m6**: About auxiliary properties of black-box optimisation problems.
>
> **A**: We mentioned those properties but we now re-emphasize them in Section 2.1.
>
> **m7**: I would recommend using a different Greek letter for the design space since it bears a resemblance to the symbol for the oracle.
>
> **A**: Thanks for the advice. We have changed the letter to a calligraphic M.
>
> **m8**: 2D molecular graph representations and stereo chemistry.
>
> **A**: We have added a footnote to explain that there are ways of augmenting graph representations to capture stereochemistry
>
> **m9**: Removing “exhaustively” in Screening.
>
> **A**: We have revised the sentence accordingly.
>
> **m10**: In terms of the use of MolPAL, it may be worth mentioning that many other choices of surrogates could be used.
>
> **A**: Thanks for the advice. We have added more references about machine learning accelerated virtual screening when introducing this concept.
>
> **m11**: Worth citing the poor scaling of the SSK kernel
>
> **A**: We have added the poor scaling about the SSK.
>
> **m12**: Reference 26, Auto-encoding variational Bayes was published at ICLR 2014.
>
> **A**: From the first-hand evidence that we find (https://openreview.net/forum?id=33X9fd2-9FyZd&source=post_page---------------------------), no decision was made for this paper in ICLR 2014. We cannot find any publication record for this paper elsewhere, so we are keeping our original reference. If you can find a publication record, we are happy to change it.
>
> **m13 & 15**: It would be great if BoTorch and Adam were cited.
>
> **A**: We’ve added them.
>
> **m14**: Poor scaling is due to using exact GP implementations.
>
> **A**: We agree that changing exact GPs to sparse GP might make the methods more scalable, but as we mentioned in the response to **M3**, the early stop doesn’t have a large influence on the VAE+BO methods. For GP BO, the exact GP model is adopted from its original implementation [4]. To avoid misunderstanding, we have revised the sentence.
>
> **m16 & 17**: Typo, "determinantal point process" and missing capitalisation.
>
> **A**: We’ve fixed them.
>
> Reference
> ===
>
> [1] https://www.rdkit.org/docs/GettingStartedInPython.html#list-of-available-fingerprints
>
> [2] Thawani et al. The photoswitch dataset: a molecular machine learning benchmark for the advancement of synthetic chemistry. arXiv 2020.
>
> [3] Capecchi et al. One molecular fingerprint to rule them all: drugs, biomolecules, and the metabolome. Journal of cheminformatics 2020.
>
> [4] https://github.com/AustinT/ai4sci-2021-denovo-benchmarks

---

### Meta-Review · Area_Chair_HML6 · 2022-09-03

**Recommendation:** Accept
**Confidence:** 4

**Metareview:**

Five out of the six reviewers are supportive to this work. There have been very extensive discussions between the authors and reviewers during rebuttals, especially the one reviewer who provided slightly negative recommendations. Overall, I feel most of the concerns have been addressed and thus recommend an accept.

---

### Decision · Program_Chairs · 2022-09-16

Accept